# A Pragmatic Note on Evaluating Generative Models with Fréchet Inception Distance for Retinal Image Synthesis

**Yuli Wu**[1,3]  MAIL@WUYULI.COM

**Fucheng Liu**[1]  FUCHENG.LIU@RWTH-AACHEN.DE

**Rüveyda Yilmaz**[1]  RUEVEYDA.YILMAZ@LFB.RWTH-AACHEN.DE

**Henning Konermann**[1]  HENNING.KONERMANN@LFB.RWTH-AACHEN.DE

**Peter Walter**[1,2]  PWALTER@UKAACHEN.DE

**Johannes Stegmaier**[1,3]  JOHANNES.STEGMAIER@HHU.DE

[1] *RWTH Aachen University, Aachen, Germany*

[2] *Uniklinik RWTH Aachen, Aachen, Germany*

[3] *Heinrich Heine University Düsseldorf, Düsseldorf, Germany*

**Editors:** Accepted for publication at MIDL 2026

## Abstract

Fréchet Inception Distance (FID), computed with an ImageNet pretrained Inception-v3 network, is widely used as a state-of-the-art evaluation metric for generative models. It assumes that feature vectors from Inception-v3 follow a multivariate Gaussian distribution and calculates the 2-Wasserstein distance based on their means and covariances. While FID effectively measures how closely synthetic data match real data in many image synthesis tasks, the primary goal in biomedical generative models is often to enrich training datasets ideally with corresponding annotations. For this purpose, the gold standard for evaluating generative models is to incorporate synthetic data into downstream task training, such as classification and segmentation, to pragmatically assess its performance. In this paper, we examine cases from retinal imaging modalities, including color fundus photography and optical coherence tomography, where FID and its related metrics misalign with task-specific evaluation goals in classification and segmentation. We highlight the limitations of using various metrics, represented by FID and its variants, as evaluation criteria for these applications and address their potential caveats in broader biomedical imaging modalities and downstream tasks.

**Keywords:** FID, Generative models, Retinal imaging.

## 1. Introduction

Deep generative models, particularly generative adversarial networks (GANs) (Goodfellow et al., 2014), by adversarial training of the generator and discriminator, and diffusion models (Ho et al., 2020), by iteratively refining noise into structured images, have demonstrated significant potential in 2D and 3D biomedical image synthesis, as shown in studies such as Eschweiler et al. (2024); Han et al. (2020); Müller-Franzes et al. (2023); Wu et al. (2024); Yilmaz et al. (2024, 2025). By learning from real biomedical data, these models can generate realistic synthetic images, helping to address challenges like limited data availability and data privacy concerns. Correspondingly, various studies have focused on generating fully annotated images to support downstream task training, such as classification and segmentation (Park et al., 2019; Zhang et al., 2023). This can be achieved by conditioning

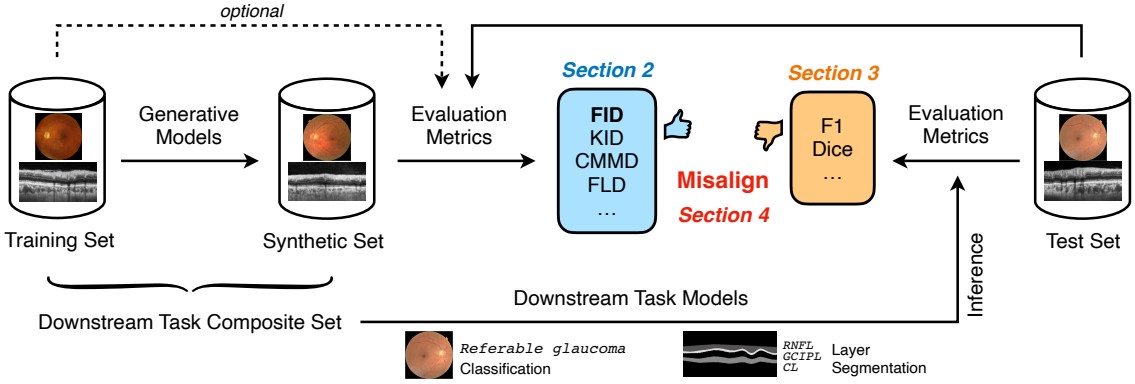

Figure 1: Overview of the generative model pipeline for data enrichment exemplified with retinal images, including generation, evaluation, and downstream task prediction. The discrepancy between generative evaluation metrics and downstream performance is investigated.

generative models on different types of input, such as text, images, or sketches. Furthermore, several approaches control the diffusion process using structured guidance, where segmentation maps are used to steer image synthesis (Eschweiler et al., 2024; Park et al., 2019; Yilmaz et al., 2024; Zhang et al., 2023). The resulting image-annotation pairs are then pragmatically used for training downstream tasks. However, ensuring that synthetic images are both realistic and useful for downstream tasks remains a critical challenge (Theis et al., 2016; Konz et al., 2024a).

In ophthalmic imaging particularly, generative models have enabled explainable classification with GAN-based styles (Lang et al., 2021), lesion-map conditioned diabetic retinopathy fundus generation using GANs (Hou, 2023), diffusion-based fundus image generation (Müller-Franzes et al., 2023), OCT image–label generation for layer segmentation (Wu et al., 2024; Huang et al., 2024), counterfactual retinal synthesis (Ilanchezian et al., 2025), and controllable fundus synthesis via disentangled representations (Müller et al., 2025).

A key open question is whether *feature-distance* metrics (see section 2), which quantify distributional similarity in the embedding space of a fixed pretrained model, are reliable proxies for the practical utility of synthetic data in downstream training. In this work, we study data enrichment and ask: (i) do different feature-distance metrics yield consistent rankings of generative model variants, as quantified by Kendall's $\tau$ rank correlation; and (ii) do these rankings align with downstream performance in classification and segmentation when synthetic samples are added to the training set? To answer these questions, we evaluate multiple generators on retinal fundus and OCT synthesis, compute seven representative metrics spanning different distances and feature extractors, and compare their rank orderings with downstream task performance on held-out test data.

As shown in Figure 1, we illustrate a common scenario using generative models, demonstrated with two retinal image modalities for data enrichment. On one hand, a synthetic dataset is generated with optional annotations and evaluated using various metrics, primarily FID and its variants (Section 2). On the other hand, the composite dataset containing a

mixture of real and synthetic data is applied to a downstream task, where the trained model predicts the desired outputs on the test set (Section 3). In Section 4, we demonstrate that widely used generative evaluation metrics fail to align with downstream performance and draw a pragmatic note on the unreliability of such *feature-distance* metrics when assessing generative models for data enrichment in the context of a utilitarian downstream task.

## 2. Generative Evaluation Metrics

### 2.1. Fréchet Inception Distance

Fréchet Inception Distance (FID) (Fréchet, 1957; Heusel et al., 2017) is the de facto standard metric for assessing the perceptual quality of generated images. It compares the feature distributions of generated and real images using an ImageNet (Deng et al., 2009) pretrained Inception-v3 model (Szegedy et al., 2016) and the Fréchet distance (equivalent to the 2-Wasserstein distance (Vaserstein, 1969; Peyré and Cuturi, 2019)), which measures the difference between two probability distributions: $p_r$, the distribution of real data, and $p_g$, the distribution of generated data. Assuming Gaussian distributions with means $\mu_r, \mu_g$ and covariances $\Sigma_r, \Sigma_g$, the squared Fréchet distance (FD) is derived as:

$$D^2_{\text{Fréchet}}(p_r, p_g) = \|\mu_r - \mu_g\|^2 + \text{tr}\left(\Sigma_r + \Sigma_g - 2(\Sigma_r \Sigma_g)^{1/2}\right), \tag{1}$$

where $\|\mu_r - \mu_g\|^2$ is the squared Euclidean distance between means, and $\text{tr}(\cdot)$ is the trace of a matrix. A lower FID score indicates that the generated images more closely resemble the real images in terms of perceptual similarity. It is important to note that the closed-form expression in Equation (1) is not limited to the multivariate Gaussian distribution, but also applies to any two distributions in $\mathbb{R}^n$ within the family of *elliptically contoured distributions* (Dowson and Landau, 1982; Peyré and Cuturi, 2019), expressed as:

$$f(x; \mu, A) = \text{const.} \times \frac{1}{|A|^{1/2}} \, g\left((x - \mu)^\top A^{-1}(x - \mu)\right), \tag{2}$$

where $A$ is a positive definite matrix and $g(z)$ is a non-negative function defined on the positive real axis $(r)$, satisfying $0 < \int_0^\infty r^{n/2-1} g(r) dr < \infty$, which characterizes the specific distribution (e.g., for the Gaussian distribution, $g(z) = \exp(-z/2)$). Therefore, merely verifying the non-Gaussianity of the feature distributions does not undermine the reliability of FID; however, this does not extend to Gaussian mixtures, which are generally not elliptically bounded (cf. Jayasumana et al. (2024)).

By substituting the feature extractor in the FID calculation with alternative models or applying different distance measures for feature comparison, we can explore alternative approaches to evaluate generative models. We refer to this class of generative evaluation metrics as *feature-distance* metrics and present variants of FID, categorized by different distance measures in Section 2.2 and feature extractors in Section 2.3.

### 2.2. Distance and Divergence Variants

Besides the Fréchet distance, Maximum Mean Discrepancy (MMD) (Gretton et al., 2006) is another key distance metric for comparing probability distributions, commonly used to

evaluate generative models. MMD is a non-parametric measure that compares the means of two distributions in a higher-dimensional feature space. The squared MMD is defined as:

$$D^2_{\mathrm{MMD}}(p_r, p_g) = \mathbb{E}_{x,x' \sim p_r}[k(x, x')] + \mathbb{E}_{y,y' \sim p_g}[k(y, y')] - 2\mathbb{E}_{x \sim p_r, y \sim p_g}[k(x, y)], \qquad (3)$$

where $\phi(\cdot)$ is a feature map and $k(x, y) = \langle \phi(x), \phi(y) \rangle$ is a kernel function, which can be, e.g., a rational quadratic kernel as in Kernel Inception Distance (KID) (Bińkowski et al., 2018) or a Gaussian RBF kernel as in CLIP-MMD (CMMD) (Jayasumana et al., 2024). Unlike FID, MMD does not assume a specific distribution form, making it more flexible and capable of handling a broader range of distributions.

Various studies have attempted to estimate feature distributions using Gaussian Mixture Models (GMMs), such as in the case of class-aware Fréchet distance (CAFD) (Liu et al., 2018). Similarly, Wasserstein-GMM (WaM) (Luzi et al., 2023) introduces an approximate 2-Wasserstein metric based on the fitted mixture of Gaussian (MoG) distributions for both real and generated data. Furthermore, Feature Likelihood Divergence (FLD) (Jiralerspong et al., 2023) estimates the Kullback-Leibler divergence (KLD) between learned MoG distributions, claiming to capture the novelty, fidelity, and diversity of generated samples. Although KLD is not a true distance due to its lack of symmetry and violation of the triangle inequality, it is still included in this section for the sake of completeness.

### 2.3. Feature Extractor Variants

In addition to Inception-v3 (Szegedy et al., 2016), recent advancements in foundation models, such as the vision-language model CLIP (Radford et al., 2021) and the self-supervised vision foundation model DINOv2 (Oquab et al., 2024), provide more powerful and general alternatives for generative feature extraction. Moreover, Fréchet AutoEncoder Distance (FAED) (Buzuti and Thomaz, 2023) leverages the latent features from a VQ-VAE (van den Oord et al., 2017).

As a special case of the feature extractor variants, it is natural to consider using a modality-specific model for feature extraction in biomedical generative evaluation. The expectation is that a modality-specific feature extractor would produce better features than a general model. However, as demonstrated in (Woodland et al., 2024), pretraining on a radiology image dataset RadImageNet (Mei et al., 2022) leads to a poorer correlation with human judgment of realisticity (without evaluating downstream performance) compared to a model pretrained on ImageNet (Deng et al., 2009). In this study, we adopt RETFound (Zhou et al., 2023), a foundation model pretrained on retinal images using masked autoencoders (He et al., 2022), as the feature extractor to assess the pragmatic reliability of modality-specific feature-distance metrics with respect to the downstream performance.

### 2.4. Other Related Metrics

Several other generative evaluation metrics complement FID and assess various aspects of model performance. The Peak Signal-to-Noise Ratio (PSNR) is a traditional metric that evaluates image quality based on pixel-level differences, though it does not account for perceptual aspects of image quality. To address this, the Structural Similarity Index Measure (SSIM) (Wang et al., 2004) considers luminance, contrast, and structure in its comparison, providing a more perceptually meaningful measure of similarity between generated and real

images. Another widely used metric is the Inception Score (IS) (Salimans et al., 2016), which, similarly to FID, leverages a pretrained Inception network (Szegedy et al., 2016) to assess the diversity and quality of generated images based on classification confidence, though it has been critiqued for not fully addressing issues like mode collapse. Unbiased FID (Chong and Forsyth, 2020) introduces a bias-free metric by extrapolating FID scores to an infinite sample set, and shows that Quasi-Monte Carlo integration improves the estimation of FID for finite samples. Finally, the Clean FID (Parmar et al., 2022) metric has been introduced to improve upon FID by addressing aliasing artifacts that can arise from low-level image quantization and resizing, enhancing its robustness and comparability.

### 2.5. Demonstrated Metrics

In this paper, we report seven diverse generative evaluation metrics, covering Fréchet distance (FID, Clean-FID, CLIP-FD, RETFound-FD), MMD (KID, CMMD), KLD (FLD) and feature extractors, including ImageNet pretrained Inception-v3 (FID, Clean-FID, KID), CLIP (CLIP-FD, CMMD), DINOv2 (FLD), RETFound (RETFound-FD). In our experiments, all metrics are calculated between the generated data and the unseen test data for the downstream task, except for FLD, which uses both the training and test sets. An overview of the generative evaluation metrics discussed in this section is presented in Table A1.

## 3. Experiments

In this study we selected StyleGAN3 (Goodfellow et al., 2014; Karras et al., 2021) and two diffusion-based architectures, namely Medfusion (Rombach et al., 2022; Müller-Franzes et al., 2023) and DDPM (Ho et al., 2020; Eschweiler et al., 2024; Wu et al., 2024), because GANs and diffusion models remain among the best performing and most widely used generative frameworks in both natural and biomedical imaging. These models therefore provide a representative test bed for evaluating generative metrics in realistic settings. For StyleGAN3, we obtain model variants by selecting checkpoints along training according to their validation FID, which reflects standard practice in GAN model development and mirrors how researchers typically identify high- and low-performing generations. For diffusion models, we follow the conventional procedure of varying the number of sampling steps $t$, which directly controls synthesis quality and produces systematically different generations without retraining the model. Together, these approaches yield diverse model checkpoints that differ in perceptual quality in a principled and reproducible manner, allowing us to examine how generative metrics behave across a spectrum of model performance.

### 3.1. Color Fundus Photography

**Fundus Dataset.** We utilize the AIROGS dataset (De Vente et al., 2023), which consists of approximately 101,000 color fundus images, labeled as *no referable glaucoma* (NRG) and *referable glaucoma* (RG). The dataset is split into training and test sets at an 80:20 ratio. Specifically, the training set contains 78,537 NRG and 2,616 RG images, while the test set contains 19,635 NRG and 654 RG images.

**Fundus Image Synthesis.** Two deep generative models are employed for realistic fundus image synthesis: the advanced generative adversarial network StyleGAN3 (Karras et al.,

2021) and the medical image-specific latent diffusion model (Rombach et al., 2022), Medfusion (Müller-Franzes et al., 2023). StyleGAN3 is trained only on RG fundus images to generate new RG samples. To verify that the generated images are indeed RG, we trained a binary classifier (distinguishing RG from NRG) that achieves 93.2% accuracy on the generated images. We then select ten checkpoints based on the decreasing FID against the full dataset (i.e., fid50k_full in Karras et al. (2021)), with FID values of {194, 149, 118, 87, 57, 46, 37, 25, 16, 6}, denoted as SG-1 to SG-10. Diffusion models offer a more convenient way to obtain generative models of varying qualities corresponding to the sampling steps $t$ compared to GANs with different checkpoints. For Medfusion, we select seven models with varying $t$ from {5, 10, 15, 25, 75, 150, 250}, denoted as MF-1 to MF-7. In both synthesis approaches, 75,921 RG fundus images are synthesized to supplement the imbalanced dataset. For metric evaluation, we use 6,000 synthetic images—enough to capture diversity without distorting FID due to low-rank covariance caused by outweighing the 654-image test set (Jayasumana et al., 2024; Konz et al., 2024a). Similarly, we performed an auxiliary sanity check of label consistency by training a lightweight classifier on real data and evaluating it on the synthetic fundus images generated by Medfusion, obtaining an RG classification accuracy of 94.1%.

**Fundus Downstream Task.** The downstream task involves binary classification with class imbalance, where the minor class (RG) is augmented with synthesized data. Two widely used architectures are adopted: ResNet-50 (He et al., 2016) and Swin Transformer Tiny (Swin-T) (Liu et al., 2021). The F1 score according to the referable glaucoma class is calculated to highlight the low recall for RG, with imbalanced baseline F1 scores of 64.57% for ResNet-50 and 63.73% for Swin-T (cf. Figure 2(b)).

## 3.2. Optical Coherence Tomography

**OCT Dataset.** We utilize the dataset from the MICCAI GOALS Challenge (Fang et al., 2022), consisting of 100 pixel-wise labeled circumpapillary Optical Coherence Tomography (OCT) images, split 50:50 for training and test. Three layers are annotated on the OCT scans, namely the retinal nerve fiber layer (RNFL), the ganglion cell-inner plexiform layer (GCIPL), and the choroid layer (CL). We note that this OCT setting is a small-sample regime (50 test images), which can increase the instability of FID (Jayasumana et al., 2024). This regime is nevertheless representative for many annotated biomedical segmentation datasets (e.g., Fang et al. (2022); Wu et al. (2023)), where data collection and pixel-wise annotation are expensive.

**OCT Image Synthesis.** Following Eschweiler et al. (2024) and Wu et al. (2024), we employ a denoising diffusion probabilistic model (DDPM) to generate realistic retinal OCT images with a sketch from a processed segmentation mask, which enables the generation of fully annotated synthetic data. Similarly, we select seven diffusion models according to increasing sampling steps $t$ of {100, 150, 200, 250, 300, 350, 400}, denoted as DM-1 to DM-7. Layer statistics from 50 real OCT images are applied as priors to generate sketches, producing 200 synthetic ones as the new training set. We further performed an analogous sanity check for OCT by training a lightweight segmentation model on real data and evaluating it on synthetic OCT images with their generated layer labels. The Dice score on the generated samples is 77.9%, indicating that the generated segmentation labels are

broadly plausible but not fully consistent with the predictions of a real-data-trained model (cf. Wu et al. (2024)).

**OCT Downstream Task.** We focus on the layer segmentation task using two well-performing architectures: $U^2$-Net (Qin et al., 2020) and TransUNet (Chen et al., 2024). Following Fang et al. (2022), Dice scores for the test set segmentations of three retinal layers are computed using weights of 0.4, 0.3, and 0.3 for RNFL, GCIPL, and CL, respectively.

## 4. Results

### 4.1. Feature Sparsity and Entropy

We begin with analyzing the sparsity with approximated L0 norm and the entropy of feature vectors from generated OCT images in Figure $2(a)$. At the interface between feature extraction and distance measurement, sparsity and entropy of the features help to understand low-level behaviors of the evaluation pipeline. On the raw feature vectors, we count absolute values above a threshold 0.01 and compute the relative L0 norm with respect to dimensionality. Across four feature extractors on two retinal imaging modalities, the features from DINOv2 (Oquab et al., 2024) show the lowest sparsity, while the ImageNet pretrained Inception-v3 (Deng et al., 2009; Szegedy et al., 2016) yields the most sparse feature vectors.

Moreover, entropy (in nats) is calculated on the probability vectors, derived by applying a sigmoid function and normalizing the feature vectors (sum = 1). Lower entropy suggests that the feature vectors carry less information, with more values concentrated in specific dimensions, as seen with Inception (Deng et al., 2009; Szegedy et al., 2016). In contrast, higher entropy indicates a more even distribution of information across dimensions, with CLIP (Radford et al., 2021) exhibiting the highest entropy among four models. These feature-level properties help illuminate why metrics based on these features may behave inconsistently across downstream tasks. Since entropy is computed after normalization, in a high-dimensional space (up to $d = 2048$) it concentrates near a model-specific mean for dense representations; consequently, different image sets can yield similar entropy values for the same extractor. However, entropy remains informative *across* extractors, as different architectures and pretraining objectives induce distinct feature concentration patterns, which influence the behavior of feature-distance metrics. We therefore use entropy as diagnostic statistics of representation geometry, rather than as a measure of image complexity.

Although not a full explanation, sparsity and entropy patterns demonstrate fundamental differences among feature extractors that may contribute to misalignment. An interesting observation is that the domain knowledge encoded in the retinal foundation model RETFound (Zhou et al., 2023) does not lead to feature representations with improved sparsity or entropy compared to the other pretrained extractors.

### 4.2. Consistent Trends of Metrics

Table 1 and Table 2 report seven generative evaluation metrics for two retinal modalities and downstream tasks across 24 models, including StyleGAN3 and two diffusion models. To assess the consistency between these metrics, we compute the Kendall's $\tau$ coefficient for all metric pairs, resulting in 63 pairs (21 metric pairs across 3 generative approaches; see Figure 3). Almost all of these pairs, except one, exhibit a Kendall's $\tau$ coefficient greater

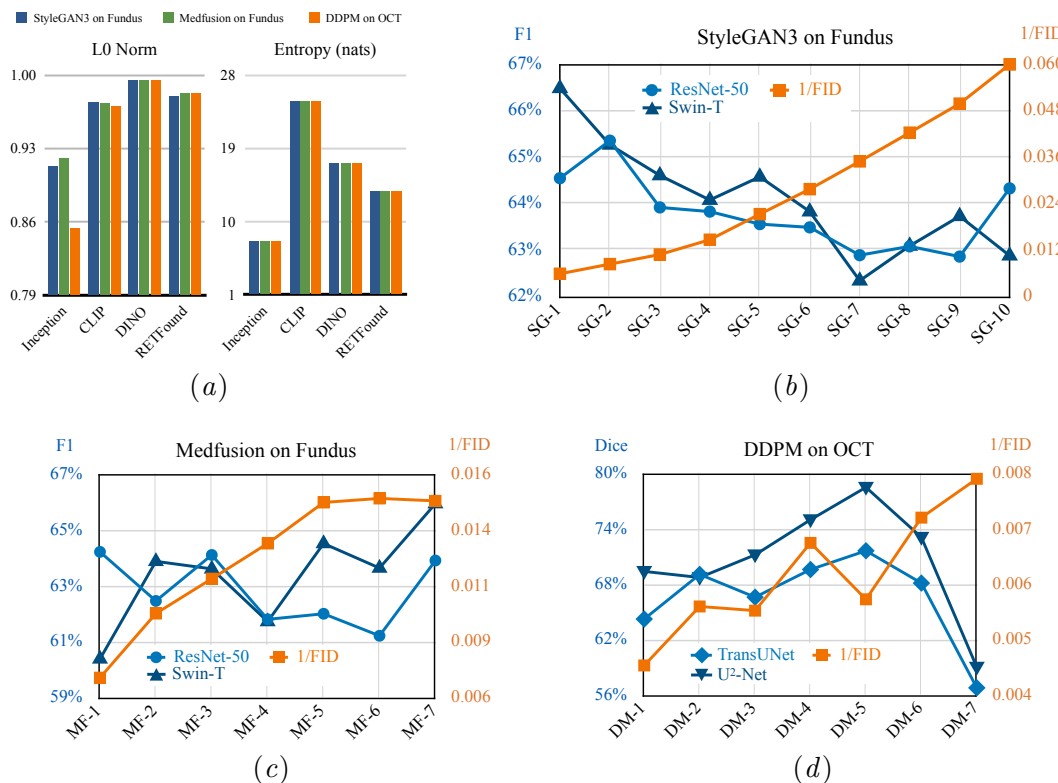

Figure 2: (a) Feature sparsity (approximated using the relative L0 norm) and entropy (in nats) for feature vectors extracted by different pretrained models. These statistics illustrate how differently each backbone represents synthetic images, which can influence the behavior of feature-distance metrics. (b)–(d) Comparison of downstream performance with the reciprocal of FID. Note that the reciprocal of FID is used in this figure solely for visualization purposes. The standard FID values and all other evaluation metrics are reported elsewhere in this paper. The left vertical axes (blue) show downstream evaluation scores, and the right vertical axes (orange) show 1/FID. In an *ideal* scenario, both curves would exhibit similar trends, indicating agreement between the perceptual metric and the downstream task performance.

than 0.5, with the majority (78%) showing a $\tau$ coefficient above 0.7, indicating a strong correlation among these *feature-distance* generative evaluation metrics, even though the features are distinct in sparsity and entropy from different extractors (Section 4.1).

As demonstrated in the upper left triangles in Figure 3, the strong internal agreement among feature-distance metrics highlights their redundancy: despite differences in feature extractors and distances, these metrics largely rank models identically, yet this ranking fails to align with downstream utility. This redundancy generalizes from FID to other generative metrics (Figure 2) and contextualizes why many proposed variants of FID offer

Table 1: Generative evaluation metrics for fundus image synthesis with StyleGAN3 (SG) (Karras et al., 2021) and Medfusion (MF) (Müller-Franzes et al., 2023). Kendall's $\tau$ coefficients and $p$-values are reported for each metric in relation to the mean F1 score of ResNet-50 (He et al., 2016) and Swin-T (Liu et al., 2021). Note that $\tau = 1$ in the worst case and $\tau = -1$ in the best case.

| Models | Fréchet Distance | | | | KID | CMMD | FLD |
|--------|-----------|-------|------|----------|-----|------|-----|
|        | Inception | Clean | CLIP | RETFound |     |      |     |
| SG-1   | 175.99 | 173.94 | 53.19 | 41.77 | 0.2073 | 5.165 | 106.45 |
| SG-2   | 121.59 | 121.15 | 34.47 | 61.02 | 0.1391 | 2.980 | 92.67 |
| SG-3   | 95.29  | 93.20  | 23.45 | 33.64 | 0.1009 | 1.975 | 80.68 |
| SG-4   | 69.70  | 68.66  | 15.88 | 33.38 | 0.0662 | 1.563 | 73.42 |
| SG-5   | 49.45  | 47.22  | 7.76  | 23.51 | 0.0402 | 0.986 | 60.07 |
| SG-6   | 39.17  | 36.17  | 5.91  | 19.14 | 0.0277 | 0.736 | 47.66 |
| SG-7   | 30.92  | 28.70  | 5.33  | 15.12 | 0.0151 | 0.619 | 35.01 |
| SG-8   | 24.83  | 23.66  | 4.81  | 14.82 | 0.0111 | 0.489 | 31.29 |
| SG-9   | 21.11  | 20.09  | 4.60  | 11.25 | 0.0089 | 0.440 | 26.56 |
| SG-10  | 17.30  | 16.69  | 4.02  | 9.26  | 0.0063 | 0.421 | 21.76 |
| $\tau_{\text{Kendall}}$ | 0.69 | 0.69 | 0.69 | 0.64 | 0.69 | 0.69 | 0.69 |
| $p_{\text{Kendall}}$ | ** | ** | ** | ** | ** | ** | ** |
| MF-1   | 148.82 | 145.51 | 37.41 | 50.05 | 0.1632 | 1.537 | 67.72 |
| MF-2   | 105.81 | 102.56 | 17.66 | 41.65 | 0.1109 | 0.735 | 53.80 |
| MF-3   | 91.28  | 88.43  | 12.53 | 38.87 | 0.0940 | 0.631 | 52.70 |
| MF-4   | 80.00  | 77.60  | 9.51  | 36.42 | 0.0823 | 0.573 | 45.86 |
| MF-5   | 68.91  | 67.91  | 6.52  | 34.40 | 0.0740 | 0.558 | 41.36 |
| MF-6   | 67.76  | 67.07  | 6.01  | 34.60 | 0.0741 | 0.562 | 42.54 |
| MF-7   | 68.00  | 67.55  | 5.94  | 35.20 | 0.0749 | 0.567 | 40.95 |
| $\tau_{\text{Kendall}}$ | -0.24 | -0.24 | -0.33 | -0.24 | -0.24 | -0.25 | -0.43 |
| $p_{\text{Kendall}}$ | n.s. | n.s. | n.s. | n.s. | n.s. | n.s. | n.s. |

limited practical improvement. In addition, FID is known to be unreliable in small-sample regimes, and several variants have been proposed to mitigate these issues (e.g., CMMD (Jayasumana et al., 2024)). However, in our experiments these variants do not exhibit meaningful improvements over vanilla FID.

## 4.3. Misaligned FID and Downstream Performance

Kendall's $\tau$ coefficients and $p$-values are provided in Table 1 and Table 2 for each generative evaluation metric (the lower the better) and their correlation with downstream task performance (F1 or Dice, the higher the better). A $\tau = 1$ indicates negative correlation between the generative model's evaluation and downstream performance, while $\tau = -1$ suggests an ideal generative evaluation metric. The results show that for diffusion models, these metrics fail to capture downstream performance, as indicated by the non-significant $p$-values (n.s. when $p \geq 0.05$). More critically, for StyleGAN3, the metrics predict performance in

Table 2: Generative evaluation metrics for optical coherence tomography synthesis with DDPM (Ho et al., 2020). Kendall's $\tau$ coefficients and $p$-values are reported for each metric in relation to the mean Dice score of U$^2$-Net (Qin et al., 2020) and TransUNet (Chen et al., 2024).

| Models | Fréchet Distance | | | | KID | CMMD | FLD |
|--------|-----------|-------|------|----------|------|------|-----|
| | Inception | Clean | CLIP | RETFound | | | |
| DM-1 | 212.67 | 232.04 | 13.15 | 54.87 | 0.2578 | 1.005 | 38.74 |
| DM-2 | 174.95 | 192.75 | 10.73 | 45.79 | 0.1925 | 0.795 | 29.65 |
| DM-3 | 177.17 | 197.43 | 10.33 | 44.08 | 0.1958 | 0.884 | 32.84 |
| DM-4 | 146.87 | 167.19 | 9.37 | 44.71 | 0.1503 | 0.972 | 22.01 |
| DM-5 | 171.27 | 184.88 | 8.81 | 33.15 | 0.1860 | 0.775 | 25.58 |
| DM-6 | 138.07 | 156.05 | 6.69 | 28.54 | 0.1390 | 0.748 | 29.56 |
| DM-7 | 126.42 | 142.63 | 6.33 | 34.32 | 0.1052 | 0.698 | 17.91 |
| $\tau_{\text{Kendall}}$ | -0.14 | -0.14 | -0.14 | -0.24 | -0.14 | -0.05 | -0.33 |
| $p_{\text{Kendall}}$ | n.s. | n.s. | n.s. | n.s. | n.s. | n.s. | n.s. |

the opposite direction with $0.001 \leq p < 0.01$, denoted as $**$. We also illustrate the downstream performance with 1/FID to better depict the relationship in Figures 2(b), 2(c) and 2(d). No clear correlation is observed across the three plots, highlighting the unreliability of FID (and, by extension, the other six metrics due to their consistency, as discussed in Section 4.2) for evaluating generative models in the context of a downstream task.

## 5. Limitations and Outlook

Although this study evaluates three generative models across two retinal imaging modalities, the scope remains limited to color fundus photography and retinal OCT. These modalities are representative for retinal research, but do not capture the diversity of biomedical imaging. Prior findings in radiology, for example by Konz et al. (2024a), suggest that similar misalignment effects may also arise in other imaging domains, although a systematic cross-modality investigation was beyond the scope of this work. That study examines radiology image-to-image translation, whereas our work focuses on retinal image synthesis using multiple generative models and downstream tasks, providing complementary evidence within a different modality setting. Furthermore, we did not investigate domain-specific metrics such as the Fréchet Radiology Distance (FRD) (Konz et al., 2024b), which incorporates handcrafted radiological features. Developing analogous retinal-specific metrics is possible, but we believe a more generic solution is needed. Finally, while we observe inverted or weak correlations between feature-distance metrics and downstream performance in some settings, we do not claim a definitive mechanism; plausible contributors include reduced diversity, underrepresentation of hard cases, and imperfect label–image consistency.

Future work should extend the analysis to additional biomedical modalities and to more recent conditional or controlled generative models, e.g., control-guided diffusion (Zhang et al., 2023), and explore evaluation strategies that do not rely on manually engineered

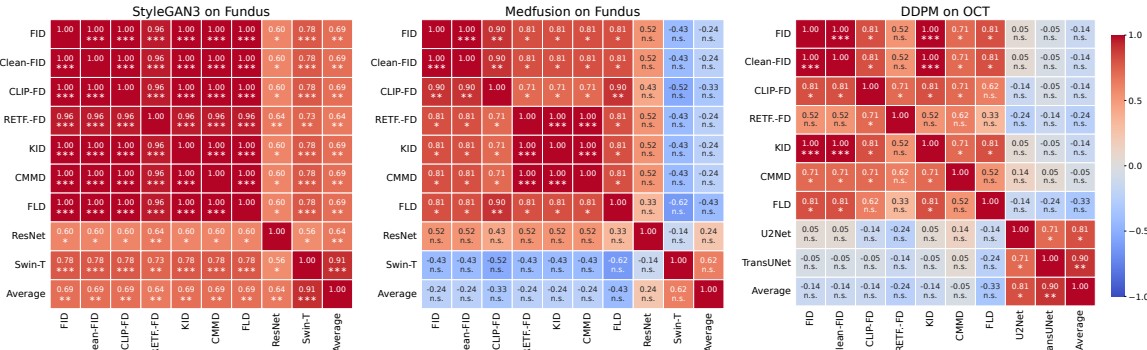

Figure 3: Kendall's $\tau$ rank correlations between feature-distance metrics and downstream task performance for all generative settings. Each cell reports the Kendall's $\tau$ value (in $[-1, 1]$) and its corresponding $p$-value (*** for $p < 0.001$, ** for $0.001 \leq p < 0.01$, * for $0.01 \leq p < 0.05$, and n.s. for $p \geq 0.05$). Warm colors denote stronger positive correlations and cold colors denote negative correlations. Average denotes the mean downstream performance across the two classification or segmentation models. The results indicate that (i) newer generations of feature-distance metrics do not consistently improve evaluation quality compared to classical variants, and (ii) feature-distance metrics rarely align with downstream performance and often exhibit inverted or insignificant correlations.

domain features, aiming instead for more general and theoretically grounded metrics. Since our findings indicate that downstream evaluation provides the most reliable measure of generative model utility, an important challenge is to integrate such task-based assessment into model selection workflows without incurring prohibitive computational cost. Approaches such as Bayesian optimization, surrogate modeling, or partial downstream evaluation could help make downstream-aware generative model selection practical at scale.

## 6. Conclusion

In this study, we evaluated three generative models across two retinal imaging modalities and showed that Fréchet Inception Distance (FID) and a broad set of related feature-distance metrics do not reliably reflect downstream task performance when synthetic data are used for data augmentation. Despite their widespread adoption, these metrics often fail to capture the practical usefulness of generated images for classification or segmentation. Our results therefore suggest that downstream evaluation should serve as the primary criterion when assessing generative models intended for biomedical data enrichment. Identifying proxy metrics that correlate more closely with downstream utility, while remaining computationally efficient, remains an important direction for future research.

## Acknowledgments

This work was supported by Deutsche Forschungsgemeinschaft (DFG, German Research Foundation) with the grant GRK2610: InnoRetVision (YW, HK, project number 424556709).

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

**Appendix**    Table A1: A summary of generative evaluation metrics discussed in this paper.

| Metrics | Distances / Divergences | Feature Extractors |
|---|---|---|
| FID Heusel et al. (2017) | Fréchet distance (FD) Fréchet (1957); Dowson and Landau (1982) | Inception-v3 (ImageNet) Szegedy et al. (2016); Deng et al. (2009) |
| Clean-FID Parmar et al. (2022) | Fréchet distance (FD) | Inception-v3 (ImageNet) |
| Unbiased FID Chong and Forsyth (2020) | Bias-corrected / extrapolated FD | Inception-v3 (ImageNet) |
| CLIP-FD | Fréchet distance (FD) | CLIP Radford et al. (2021) |
| RETFound-FD | Fréchet distance (FD) | RETFound Zhou et al. (2023) |
| KID Bińkowski et al. (2018) | Maximum mean discrepancy (MMD) Gretton et al. (2006) | Inception-v3 (ImageNet) |
| CMMD Jayasumana et al. (2024) | Maximum mean discrepancy (MMD) | CLIP |
| CAFD Liu et al. (2018) | Class-aware Fréchet distance Mixtures of Gaussians (MoG)-based | Typically Inception features |
| WaM Luzi et al. (2023) | Approx. 2-Wasserstein on fitted MoG | Implementation-dependent feature space |
| FLD Jiralerspong et al. (2023) | Kullback–Leibler divergence (KLD) between learned MoG | DINOv2 Oquab et al. (2024) |
| FAED Buzuti and Thomaz (2023) | Fréchet distance (FD) | VQ-VAE latent van den Oord et al. (2017) |
| PSNR | Pixelwise error (MSE-derived) | Pixels (no feature extractor) |
| SSIM Wang et al. (2004) | Structural similarity | Pixels / local statistics (no learned extractor) |
| IS Salimans et al. (2016) | KLD-based score from predicted label distribution | Inception-v3 (ImageNet) |

