# OpenReview forum: "A Pragmatic Note on Evaluating Generative Models with Fréchet Inception Distance for Retinal Image Synthesis"
_MIDL.io/2026/Conference — MIDL 2026 Poster_

### Official Review · Reviewer_sTNX · 2025-12-30

**Confidence:** 5
**Preliminary Rating:** 5
**Final Rating:** 5

**Summary:**

The paper presents a systematic empirical evaluation of Fréchet Inception Distance (FID) and its variants for assessing generative models used for augmenting training data in retinal images. The paper evaluates 3 generators across two retinal modalities using different feature extractor backbones with the effectiveness of synthetic data validated on downstream tasks. The authors show that while the generator metrics are correlated with each other, they are misaligned with the downstream performance metrics. The results suggest that commonly used metrics are unreliable proxies for downstream utility.

**Strengths:**

Strengths:
- Demonstrates that a wide range of fidelity metrics (FID, KID, CMMD) collapse to nearly identical ranking despite using different backbones and distance measures.
- Expansive catalogue of feature extractors illustrate a fundamental limitation of the FID metric.
- Shows internal agreement among fidelity metrics do not translate to alignment with downstream  tasks.
- Performance comparison across two downstream tasks, classification and segmentation.
- Provides a clear empirical counterpoint to the common assumption that improving perceptual generative metrics leads to improved downstream performance.

**Weaknesses:**

Weaknesses:
- FID and variants are computed in small-sample regimes, particularly for OCT(200 synthetic samples, 50 real samples), despite the paper citing prior work highlighting poor sample complexity and instability of FID.
- The selected generators are dated, limiting generality of the conclusions with respect to the current state-of-the-art.
- No code provided
- Some figure captions are overly verbose and reduce readability.

**Detailed Comments:**

- The authors should explicitly discuss the FID instability in small-sample regimes and acknowledge it as a likely limitation of their work.
- Given the authors’ detailed knowledge of FID and distance-based metrics, the conclusions would benefit from a discussion of plausible directions for designing more relevant feature representations, beyond the radiomic distance discussion.
- The authors should streamline figure captions, and move interpretation of the results to the appropriate section in the text.

**Justification Of Final Rating:**

The authors have satisfactorily addressed all points raised in the initial review. The rebuttal clarifies key limitations, particularly with respect to small-sample evaluation regimes, generator choice, and reproducibility, and appropriately scopes the conclusions of the work. The additional discussion strengthens the empirical contributions and reinforces the validity of the paper’s central claims within the stated experimental scope.

**Justification Of The Preliminary Rating:**

The paper provides strong and consistent empirical results that illustrate the limitations of widely used metrics for assessing the quality of generated medical images. While there is substantial evidence in the generative AI literature indicating a general disconnect between automated quality metrics and clinical judgement, the authors present a structured evaluation paradigm that systematically demonstrates that favourable fidelity metric rankings do not translate into downstream task utility.

**Questions To Address In The Rebuttal:**

How do the authors reconcile their evaluation regime with cited work indicating poor FID sample complexity in small-data settings?
Do the authors expect their conclusions will hold for more recent, conditional and controlled generative models?

---

> ### Author Response · Authors · 2026-01-22
>
> We thank the reviewer for the positive feedback and valuable suggestions, and address the **W**eaknesses, **C**omments, and **Q**uestions as follows:
>
> **[W1, C1, Q1] Small-sample regime.**
> We acknowledge the limitation of small-sample regimes and the potential instability of FID in this setting. However, in real-world biomedical applications, tasks with a limited amount of annotated data are exactly the common use-case scenarios. In addition, some metrics proposed or advocated as improvements (e.g., CMMD [1]) also fail to align with downstream performance in our experiments. For both reasons, we draw the same practical conclusion: FID (and related feature-distance metrics) is not always trustworthy for selecting generative models for data enrichment, and downstream-task validation remains necessary. We have added an explicit discussion in **Section 3.2** regarding the size of the dataset of OCT and in **Section 4.2** when we discuss the similar behaviors across FID variants.
>
> **[W2, Q2] Dated generators.**
> We agree that StyleGAN3 and DDPM are not the latest models. However, they remain widely used baselines and still appear in common benchmark discussions (e.g., FFHQ). We acknowledge the suggestion to include more recent controlled/conditional models (e.g., ControlNet) and have added this limitation in **Section 5**. On the other hand, if a dated generator can already yield misleading conclusions under FID and its variants, then using more advanced generators does not resolve the *intrinsic limitations* of FID. Nevertheless, extending our analysis to newer controlled generators could be an interesting direction to further test the alignment between feature-distance metrics and downstream utility and we appreciate for the suggestion.
>
> **[W3] Code.**
> We did not propose a novel algorithm and rely heavily on existing codebases; nevertheless, we agree that releasing code improves reproducibility. We will share the trained checkpoints, generated images, and auxiliary evaluation scripts soon. We apologize for not including them in the initial submission.
>
> **[W4] Verbose captions.**
> We aimed to keep figure captions self-contained, even if this makes them read verbose, because readers often jump directly to figures.
>
> **[C2] Plausible feature representations.**
> We appreciate this suggestion. To be honest, prior to the experiments, we had expected that the in-domain feature extractor RETFound would provide more suitable features for evaluating generated retinal images, but our results did not support this assumption. While we will continue to explore better feature representations and distances, we also emphasize a pragmatic workaround: making downstream-aware selection more efficient via Bayesian optimization, surrogate modeling, or partial downstream evaluation, analogous to hyperparameter tuning (Section 5).
>
> [1] Jayasumana et al., “Rethinking FID: Towards a better evaluation metric for image generation,” CVPR 2024.

---

> > ### Comment · Reviewer_sTNX · 2026-01-29
> >
> > Thank you for the rebuttal. I believe the additions strengthen the paper’s contributions and appropriately scope them within the identified limitations.

---

### Official Review · Reviewer_v2jw · 2026-01-07

**Confidence:** 4
**Preliminary Rating:** 4
**Final Rating:** 5

**Summary:**

In this paper, they analyze how the Frechet Inception Distance (FID) and related metrics correlate with task-specific evaluations when synthetic data is incorporated into training for downstream tasks for retinal image modalities. They show that all the different perception metrics correlate strongly and are therefore redundant. Furthermore, their analysis found no clear correlation between the perceptual metrics and performance on downstream tasks.

**Strengths:**

- Relevance: FID is a commonly used perceptual metric for generative image models, including biomedical generative models, to incorporate synthetic data into downstream analyses. However, until now, the agreement between perceptual metrics and task performance has been unclear or insufficiently investigated.
- Experiments: Extensive analysis of seven perceptual metrics. Four variants of Frechet distance and three additional distance metrics (KID, CMMD, FLD). Each was evaluated using three generative models of two types (GAN: StyleGAN; diffusion models: Medfusion, DDPM) with checkpoints of varying perceptual quality.
- Style: Well-structured work that is easy to follow, with clean figures and tables.
- Overview of generative evaluation metrics: Detailed and formal introduction to all compared perception metrics and beyond.

**Weaknesses:**

- Feature sparsity analysis is only performed on OCT images generated with DDPM.
- The analysis is limited to retinal imaging modalities. Therefore, it is unclear whether the trends shown also apply to other medical imaging modalities.
- The article offers insights by analyzing the limitations of existing perception metrics rather than providing a solution to the problem.

**Detailed Comments:**

- Section 4.2: The following sentence should be clarified: "To assess the consistency between these metrics, we compute the Kendall’s $\tau$ coefficient for all metric pairs, resulting in 63 pairs (21 metric pairs across 3 generative approaches; see Figure 3)." Figure 3 shows three $10 \times 10$ confusion matrices, each containing a subgrid with $7 \times 7$ perceptual metric pairs. Of the 49 pairs in the $7 \times 7$ grid, the authors only had to compute 21 **unique pairs** for three model variants, correct?
- The tables should mention the feature extractors for KID (Inception-v3), CMMD (CLIP), and FLD (DINOv2). This information is only mentioned once in the text. In comparison, it is very clear with the Frechet distance variants, as a different feature extractor here means a different variant.

**Justification Of Final Rating:**

In the rebuttal, the authors adequately addressed my concerns as well as those of the other reviewers, resulting in a revised manuscript that is clearer and overall stronger. The subsequent discussion resolved my remaining questions and helped clarify the strengths and significance of the work -- particularly its contribution in highlighting important limitations of existing image quality metrics for generative models in medical imaging. While the paper does not propose a direct solution, it raises an issue that the community needs to acknowledge and discuss.

**Justification Of The Preliminary Rating:**

The paper addresses a very relevant problem in this field: selecting a generative image model based on a perception metric such as FID may not be the best model for downstream tasks. However, the paper does not offer a solution to the problem and cannot make any statements about medical image modalities that differ from retinal fundus images or OCT.

**Questions To Address In The Rebuttal:**

Please address my points in the sections "Weaknesses" and "Detailed Comments".

---

> ### Author Response · Authors · 2026-01-22
>
> We thank the reviewer for the encouraging feedback and valuable suggestions, and address the **W**eaknesses and **C**omments as follows:
>
> **[W1] Feature sparsity on other datasets.**
> We thank the reviewer for pointing out this weakness. We have added the feature sparsity and entropy analysis for the other two generated datasets as well in **Figure 2(a)** and updated the corresponding description in **Section 4.1**. Note that entropy is computed after normalization, so in $d=2048$ dimensions it concentrates near an extractor-specific mean for dense features; thus, different image sets can yield similar entropy values for the same extractor. However, entropy remains informative *across* extractors because different architectures/objectives induce systematically different feature concentration patterns (sparse vs. spread), which affects how feature-distance metrics behave. We therefore use entropy/sparsity as diagnostic statistics of representation geometry, not as a proxy for semantic image complexity. For the detailed entropy values (Figure 2(a) right):
>
> | Model / Dataset     | Inception | CLIP     | DINO     | RETFound |
> |---------------------|-----------|----------|----------|----------|
> | StyleGAN on Fundus  | 7.61723   | 24.87671 | 17.20185 | 13.79286 |
> | Medfusion on Fundus | 7.61789   | 24.87687 | 17.19826 | 13.78085 |
> | DDPM on OCT         | 7.61531   | 24.88248 | 17.17744 | 13.79820 |
>
> **[W2] Limited modalities.**
> We agree that this study is limited to retinal images and have emphasized this in the title. While we do not claim the observed misalignment generalizes universally, we still call for caution and welcome validation by the community on additional modalities.
>
> **[W3] Insights rather than a new solution/metric.**
> We acknowledge and openly state that we have not proposed a novel evaluation metric to replace FID. Our pragmatic recommendation is to directly evaluate downstream task performance when synthetic data are intended for data enrichment.
>
> **[C1] Unique pairs in Figure 3.**
> Yes, this is correct. To be more clarified: each $7\times 7$ sub-grid is symmetric and the diagonal entries are always 1, so $(7\times 7 - 7)/2 = 21$ metric pairs are unique. Across 3 generative settings, this yields 63 unique pairs.
>
> **[C2] Feature extractors in tables.**
> We thank the reviewer for the suggestion. We attempted to include both distances and feature extractors in the tables, but the page width makes this impractical. Following this and another reviewer’s suggestion, we have included a summary table in the Appendix listing all discussed metrics together with their distances/divergences and feature extractors.

---

> ### Comment · Reviewer_v2jw · 2026-01-28
>
> - Figure 2 (a): I thank the authors for adding the additional features of sparsity and entropy analysis. The authors also provide a meaningful explanation of why entropy is only meaningful across extractors. The same seems to apply to the L0 norm, although not as drastically as for entropy. Is the conclusion, therefore, that the feature extractor seems to be more important for feature quality than the underlying image generation model (at least for the three models analyzed here)? And can you explain why DDPM on OCT with Inception as the extractor seems to deviate from the other two models in terms of the L0 norm?
> - Introduction/related work: I agree with reviewer cwCN regarding the lack of clarity, in my opinion also with regard to the lack of related work on generative image models for color fundus photography (CFP) and optical coherence tomography (OCT), also to emphasize the focus on these modalities even more. Some examples are:
>     - Diffusion models for CFP and OCT: Ilanchezian, Indu; Development and validation of an AI algorithm to generate realistic and meaningful counterfactuals for retinal imaging based on diffusion models
>     - StyleGAN for CFP: Hou, Benjamin; High-fidelity diabetic retina fundus image synthesis from freestyle lesion maps, Müller, Sarah; Disentangling representations of retinal images with generative models
>     - Medfusion for CFP: Müller-Franzes, Gustav; A multimodal comparison of latent denoising diffusion probabilistic models and generative adversarial networks for medical image synthesis

---

> > ### Author Response · Authors · 2026-01-29
> >
> > > 1a. "Is the conclusion, therefore, that the feature extractor seems to be more important for feature quality than the underlying image generation model (at least for the three models analyzed here)?"
> >
> > This conclusion can be suggested, but only under additional conditions and with caveats: (i) The dataset is mono-modal (all-CFP or all-OCT). The generative models thus have much more narrow and in-domain knowledge than the feature extractors; (ii) Sparsity and entropy provide only one perspective on "feature quality" and are sensitive to implementation details such as normalization, as well as to the pretraining data/objectives of the extractors. From another perspective of "feature quality" as reflected by FID, the image generation model still plays an important role, as evidenced by the clear trends across checkpoints in Figure 2(b–d).
> >
> >
> > > 1b. "why DDPM on OCT with Inception as the extractor seems to deviate from the other two models in terms of the L0 norm?"
> >
> > This is an interesting observation. A plausible explanation might be that the ImageNet-pretrained Inception has limited knowledge of grayscale inputs compared to large foundation models (CLIP, DINO) or in-domain models (RETFound). Although grayscale OCT images are fed as three identical channels, the lack of relevant pretraining exposure may lead Inception to produce representations with different activation patterns (and thus different apparent sparsity), which could explain the deviation relative and overall disadvantage to the other extractors w.r.t. sparsity and entropy.
> >
> >
> > > 2. " ... the lack of related work on generative image models for CFP and OCT..."
> >
> > We did not intend this work to be framed as an "ophthalmic GenAI" paper, rather as an "FID critique". But we do agree that adding a brief ophthalmic generative model context improves clarity and emphasizes our focus on CFP and OCT. We will therefore add the following part to **Section 1** in the camera-ready version:
> >
> > *In ophthalmic imaging particularly, generative models have enabled explainable classification with GAN-based styles \cite{lang2021explaining}, lesion-map conditioned diabetic retinopathy fundus generation using GANs \cite{hou2023high}, diffusion-based fundus image generation \cite{muller2023multimodal}, OCT image--label generation for layer segmentation \cite{wu2024retinal,huang2024diverse}, counterfactual retinal synthesis \cite{ilanchezian2025development}, and controllable fundus synthesis via disentangled representations \cite{muller2025disentangling}.*

---

### Official Review · Reviewer_cwCN · 2026-01-09

**Confidence:** 5
**Preliminary Rating:** 4
**Final Rating:** 4

**Summary:**

FID score is often used to compare generation quality. However, given that FID relies on the Inception v3 model pre-trained on general-domain images rather than medical images, it raises the question of whether it is an appropriate metric for evaluating generated medical images. In this work, the authors provide an example using retinal image generation, where they show that the FID score is misaligned with downstream task performance, presenting a concrete case in which the use of FID as a measure of synthetic image quality should be questioned.

**Strengths:**

1. The paper is clearly motivated and the experimental design is reasonable.
2. The related work is very thorough and well organized.
3. The experiments are comprehensive and presented clearly, with well-structured analyses.
4. Overall, the quality of the work is above average, and the presentation is clear and adequate.
5. The figures are informative and easy to understand, and they effectively help readers grasp the context and key ideas.

**Weaknesses:**

1. Clarity of the introduction: The introduction is currently not clear enough and appears somewhat problematic. The first paragraph mainly states that generative AI performs well for medical data but lacks appropriate measurements of synthetic quality for downstream tasks, while the second paragraph largely describes the structure of the paper. However, the most important elements, such as the motivation of this work, the research questions being asked, and how the paper aims to address them, are not clearly articulated. In fact, the abstract provides a clearer overview of these aspects than the introduction. I believe the paper would benefit significantly from a restructured and rewritten introduction.

(from the second, are the questions that I have, and I would like the authors to answer in rebuttal:)


2. From Figure 3, it appears that regardless of the distance metric used, downstream task performance is consistently negatively correlated with generative quality. However, I did not find a discussion of this observation in the paper. Could the authors share their hypothesis or interpretation of this behavior?


3. Based on my understanding of Figure 1, the generation process produces paired data, including both images and their corresponding labels. Is there any evaluation or validation of the correctness of the generated labels? The observed misalignment might stem from multiple factors, one of which could be that, when the FID score is low, the generated labels are less accurate.

**Detailed Comments:**

1. Citation formatting is sometimes incorrect (e.g., in the Introduction, “as shown in studies such as (citations)”).
2. In Section 2.1, the use of bold text is unclear—why is this part emphasized? Is this conclusion particularly interesting in the given context? How does this non-Gaussianity matter for the rest of the text?
3. You might consider converting Section 2.2 into a table. For example, different distance measurements could be categorized based on how they are computed and presented in a table, which would make the differences easier to see.
4. Tables 1 and 2 are quite difficult to read—would it be possible to present the results using some form of chart instead?

**Justification Of Final Rating:**

I appreciate the efforts the authors made during rebuttal; I believe this paper should be accepted and even recommend as oral, as it brings a powerful case of evaluating FID score for medical cases with downstream task.

**Justification Of The Preliminary Rating:**

I think that, overall, this paper is well presented. I appreciate the motivation of the work and the perspective of not only examining the generated images but also considering downstream tasks.
However, the introduction needs to be clearer and should include proper motivation, background, research questions, and an explanation of how the experiments are set up. Another concern is that some questions remain unclear (as listed in the weaknesses). I would consider raising my score if these questions are addressed.

**Questions To Address In The Rebuttal:**

(repeat from weakness)
1. From Figure 3, it appears that regardless of the distance metric used, downstream task performance is consistently negatively correlated with generative quality. However, I did not find a discussion of this observation in the paper. Could the authors share their hypothesis or interpretation of this behavior?
2. Based on my understanding of Figure 1, the generation process produces paired data, including both images and their corresponding labels. Is there any evaluation or validation of the correctness of the generated labels? The observed misalignment might stem from multiple factors, one of which could be that, when the FID score is low, the generated labels are less accurate.

---

> ### Author Response · Authors · 2026-01-22
>
> We thank the reviewer for the insightful feedback and valuable suggestions, and address the **W**eaknesses and **C**omments as follows:
>
> **[W1] Clarity of the introduction.**
> We thank the reviewer for the suggestion to restructure the Introduction. We have added an additional paragraph in **Sec. 1** to better articulate the motivation, the research questions, and how the paper addresses them.
>
> **[W2] Apparent negative correlation / interpretation of Fig. 3.**
> In the sub-grids of Fig. 3 (metric vs. downstream performance), the *ideal* scenario corresponds to Kendall’s $\tau=-1$, because the generative metrics are *lower is better* whereas downstream metrics are *higher is better*. Across our settings, we observe a mix of inverted, weak, and non-significant correlations rather than a single consistent pattern: the StyleGAN3–fundus case is the most pronounced, followed by DDPM–OCT (where U$^2$-Net and TransUNet behave similarly), while Medfusion–fundus shows weaker and model-dependent effects (Swin-T differs from ResNet). We reported this in Sec. 4.3.
>
> We currently view several factors as plausible contributors, but we do not claim conclusive evidence for any single mechanism:
>
> (a) **Diversity / global coverage.** Improving perceptual quality may reduce diversity if sampling concentrates on a “comfort zone” of the generator. Diversity is difficult to quantify reliably; we therefore report representation diagnostics (sparsity/entropy in Fig. 2(a)), but we do not find a simple one-to-one explanation of the observed correlation patterns.
>
> (b) **Hard examples / local coverage.** A generator that optimizes perceptual similarity may underrepresent rare or ambiguous cases that are disproportionately useful for training robust downstream models.
>
> (c) **Label-image misalignment / annotation quality (see [W3] for details).** In conditional/paired generation, label misalignment can occur and may affect downstream utility even when images look realistic. We thank the reviewer for this insightful feedback.
>
> We have addressed [W2] in **Sec. 5**.
>
> **[W3] Validation of generated labels.**
> We agree with the reviewer that label correctness is an important factor. We therefore performed an auxiliary sanity check by training a lightweight downstream model on real data and using its predictions on generated images as a proxy assessment of label consistency. For fundus images, the RG classification accuracy on generated samples is 93.2% for StyleGAN3 and 94.1% for Medfusion. For OCT, the layer segmentation Dice score on generated images is 77.9%, indicating that the generated labels are generally plausible but not perfect relative to real-data-trained models. We have added this to **Sec. 3.1** and **Sec. 3.2**.
> We also note that this proxy validation is imperfect: (i) it depends on the downstream model itself and may bias the assessment toward patterns present in the real data, and (ii) label quality alone may not fully explain utility, because even imperfect synthetic labels can sometimes act as a regularizer. In an *extreme utilitarian* point of view, wrong annotations (even anatomically wrong images) are fine if they help to improve the downstream performance.
>
> **[C1] Citation formatting.**
> We thank the reviewer for noting this. We have corrected citation formatting (e.g., using `\citet{}` for in-text citations instead of the default `\cite{}` and avoiding parenthetical-only citation blocks where inappropriate).
>
> **[C2] Non-Gaussianity.**
> We agree that the emphasis was unclear. FID is often motivated via a Gaussian assumption, and prior work (e.g., [1]) discusses non-Gaussianity as a potential concern. After revisiting the literature (e.g., [2,3]), we clarify that the closed-form expression of the 2-Wasserstein distance used by FID applies beyond Gaussians to a broader family of elliptically contoured distributions. We have removed the bold formatting as this is not our main takeaway message and added the clarifying reference, while keeping the focus of the paper on downstream alignment rather than normality testing.
>
> **[C3] Converting Sec. 2.2 into a table.**
> This is a helpful suggestion. We have added an Appendix summary table that categorizes the discussed metrics by distance/divergence and feature extractor.
>
> **[C4] Tables are difficult to read.**
> We tried to improve readability by right-aligning numeric entries to make trends easier to scan. For higher-level visualization, readers can refer to **Fig. 2** for trend curves (we show FID for clarity, as variants behave similarly) and **Fig. 3** for the correlation overview via heatmaps.
>
> [1] Jayasumana et al., “Rethinking FID: Towards a better evaluation metric for image generation,” CVPR 2024.
> [2] Dowson and Landau, “The Fréchet distance between multivariate normal distributions,” Journal of Multivariate Analysis 1982.
> [3] Peyré and Cuturi, “Computational Optimal Transport with Applications to Data Sciences,” Foundations and Trends in Machine Learning 2019.

---

> > ### Comment · Reviewer_cwCN · 2026-01-28
> >
> > Very interesting. I appreciate the reply from the authors. I think the authors cleared all my concerns.
> >
> > just out of curiosity, the authors mentioned "In an extreme utilitarian point of view, wrong annotations (even anatomically wrong images) are fine if they help to improve the downstream performance." -->Do you know of any relevant literature about this point? (only for discussion, not a weakness point)

---

> > > ### Author Response · Authors · 2026-01-29
> > >
> > > Thanks a lot for your interest. We are not aware of very on-topic literatures on biomedical generative tasks. One closely related line of work would be *dataset distillation*, originating from [1], where the synthesized distilled images can look visually implausible yet still train models effectively (see Figures 2 & 3, p. 8: https://arxiv.org/pdf/1811.10959). A later work [2] also provides visualizations of the distilled samples on both methods [1,2] (see Figures F7 & F8, pp. 17 & 19, in their appendix: https://arxiv.org/pdf/2006.05929). More recent papers can be found with keywords "dataset distillation/condensation".
> > >
> > >
> > > [1] Wang et al. “Dataset distillation.” arXiv:1811.10959, 2018.
> > >
> > > [2] Zhao et al. “Dataset condensation with gradient matching.” ICLR, 2021.

---

> > > > ### Comment · Reviewer_cwCN · 2026-01-31
> > > >
> > > > Interesting! Thanks a lot:)

---

### Author Rebuttal · Authors · 2026-01-22

**Rebuttal:**

We have revised the manuscript accordingly, and all changes in the updated version are marked in red for ease of review. Please find in the supporting material below.

**Supporting Material:**

/attachment/bb6f2e528f36d7cb0ccc09f10c399df70aa90117.pdf

---

### Comment · Area_Chair_NhbM · 2026-01-27
**Engage in discussion and Update score**

Dear Reviewers,

please have a thorough look at the responses by the authors. Please acknowledge the responses and engage in discussion if anything remains unclear. Please update your final rating by clicking “Edit” → “Official Review” and providing the Final Rating by February 1st 2026 (23:59 AoE).

Best
AC

---

### Meta-Review · Area_Chair_NhbM · 2026-02-03

**Recommendation:** Accept (Oral)
**Confidence:** 5

**Metareview:**

All reviewers agree that this is an interesting and rigorous study on the usefulness of perceptual metrices such as FID for evaluating generative models in ophthalmology.

---

### Decision · Program_Chairs · 2026-02-13

Accept (Poster)